# Features of Electrostatic Fields and Their Force Action When Using Micro- and Nanosized Inter-Electrode Gaps

**DOI:** 10.3390/ma13245669

**Published:** 2020-12-11

**Authors:** Nikolai Pshchelko, Ekaterina Vodkailo

**Affiliations:** 1Department of Physics, Military Telecommunication Academy named after Budienny S. M., 194064 Saint-Petersburg, Russia; nikolsp@mail.ru; 2Department of Informatics and Computer Technology, Saint Petersburg Mining University, 199106 Saint-Petersburg, Russia

**Keywords:** electric field, discrete charge, electric adhesion

## Abstract

The present work is devoted to assessing the influence of discreteness of electric charge distribution in the double electric layer on the characteristics of the electric fields and their force action in capacitor structures with small interelectrode gaps. Due to the fact that modern technologies often use submicron-sized interelectrode gaps, it is no longer possible to consider the electrodes uniformly charged because of the discreteness of the electric charge. The corresponding development of a mathematical and physical model for the study of a non-uniform electric field is suggested. Numerical calculations are carried out, expressions, criteria, and results that are convenient for practical evaluations are obtained. The physical and mathematical model for force characteristics of a non-uniform electric field is developed. With a sufficiently small size of the interelectrode gap, the integral force effect of discretely distributed charges can be significantly higher than with a uniform distribution of the same charge. At reasonable surface charge densities, these phenomena are usually observed at interelectrode gaps less than tenths of a micrometer.

## 1. Introduction

Capacitor structures (CS) with small interelectrode gaps are the basis for a wide range of technical devices. The principle of their operation, one way or another, is based on the use of an electric field (EF). These include:-Working structures of oxide capacitors formed by electrochemical oxidation of metal and allowing to obtain highly homogeneous submicron dielectric films over a large area;-CS with movable plates—various kinds of electrostatic sensors and activators;-Electret devices, the principle of which is based on the ability of a dielectric to accumulate and maintain electric charges for a long time [1].

The interest in technologies and devices based on EF is due to a number of their advantages, the main of which are:-Extremely low power consumption. If an electret is used as a source of electrical energy, then the device may not consume electrical energy at all during operation;-Constructive simplicity and relative cheapness of the corresponding devices. An example of such devices is electret microphones, the production of which in the world now exceeds 90% of all microphones produced;-High technical characteristics of electrostatic devices. A lightweight, low-inertia membrane is often used as one of the capacitor plates. This makes it possible to transform signals with minimal distortion, to obtain high-speed CS;-The prospects for use in MEMS technology [2]. This is due to the fact that it is at small distances between the electrodes that the electrostatic forces become significant, which makes it possible to use them in various activators. The indicated advantage especially clearly demonstrates the relevance of the topic of this work, since micro- and nanoscale gaps are widely used in MEMS devices. Electroadhesive seals (EAS)—metal-to-dielectric “gluing” formed because of the forceful action of the EF. The use of electroadhesive technologies, also known as anodic bonding, makes it possible to form a permanent connection of solid objects at temperatures significantly lower than in diffusion welding [3], and the presence of a vacuum is not necessary, and the installations used are much cheaper [4,5,6,7]. EAS can be successfully used to obtain permanent strong vacuum-tight connections of dielectrics (ceramics, glass, glass, quartz, etc.,) with metal conductors and semiconductor crystals in various combinations. It is in this gap that an EF is created, the forceful action of which leads to “gluing.” Note that the gap between the surfaces to be joined is, as a rule, due to their roughness, i.e., is in the submicron range.

One of the possible practical applications of the considered influence of the EF is the possibility of increasing the adhesion of coatings deposited to substrates, both after and during their deposition, as well as a number of other effects [8,9,10,11,12,13,14,15,16,17,18,19,20,21,22]. This technological solution has great prospects for use in connection with the need to create reliable contacts [13,14,15] in electronic devices, to create protective coatings [10,11] and for other purposes [12,22].

In [4] a theory of adhesion was proposed based on the general position of Helmholtz double electric layer (DEL) at the interface of the surfaces to be joined. Then, an electronic theory of adhesion was developed, in which the processes of the formation of DEL when joining the surfaces of dissimilar materials were studied and the contribution of DEL to the strength of the resulting “glues” was analyzed [4,5,6,7]. As a result, a new direction of research was born, associated with the possibilities of using electric fields to regulate adhesion. The term “electrical adhesion” refers to situations in which electrical phenomena can affect adhesion. In [4], a connection was experimentally found between electrical adhesion and the phenomena of triboelectricity and triboluminescence, which indicates a close relationship between electrical and adhesive phenomena.

Derjagin et al in [4] point out two main factors that can increase the adhesion:
(a)Mutual diffusion of materials to be joined. An increase in the adhesion strength of the surfaces to be joined can be ensured by the diffusion of atoms and molecules, which leads to strong adhesion of the materials to be joined. The sharp interface of the contact in the process of diffusion turns into a transition layer. This process is the basis of the known diffusion bonding method and is possible using external pressure and elevated temperatures [1,3].(b)Formation of a layer of electric charges of the opposite sign on the contact. The role of this factor in the phenomena of adhesion, in our opinion, is often underestimated. Indeed, as shown in [6,7], an increase in the electric charge density at the interface under the action of an external voltage makes it possible to obtain the pressure caused by electrostatic forces at the level used in diffusion welding (≈10^7^ Pa), and to achieve not only temporary, but also permanent connection of parts. This method of joining is called electroadhesive, and the corresponding technology is called anodic fit or anodic bonding [5].

At present, intensive development of theoretical concepts of adhesion continues, while the authors note the great influence of the electrical component of adhesion [15,16,17,18,19,20,21,22].

A feature of electrostatic forces is that they become very significant only at small distances between the electrodes. If we are talking about DEL, then its thickness is usually of the nanosized dimensions. An increase in the electric field strength in DEL due to the use of an external voltage source leads to the appearance of additional charges at its boundaries. In this case, as will be shown below, the distance between these additional charges (electrons) can be not only comparable, but also can be much greater than the DEL thickness. With this state of affairs, the concept of “uniformly charged plane” obviously loses its meaning, since charges on the plane will be distributed discretely [6]. It is clear that at large distances from such a charged plane, the fields of individual (discrete) charges will average out and the plane can be considered uniformly charged. However, in this work, we will be interested in the case of small interelectrode gaps.

Thus, the examples considered show that the use of interelectrode gaps of submicron thickness is currently in demand. At the same time, the features of the behavior of EFs in such gaps have not been sufficiently studied. Therefore, the purpose of this work is to develop a physical and mathematical model for calculating the EFs and their force action when using interelectrode gaps of submicron size, taking into account the discreteness of the charge distribution. The quantitative results that will be obtained in this case will make it possible to reveal the limits of applicability of the classical approach to solving this problem and to reveal the quantitative features of EFs at small interelectrode gaps.

## 2. Methods for Calculations

### 2.1. Calculation of Distances between Charges

Let us consider in more detail the effect of the discreteness of the distribution of electric charges on the force action of the electric field. The need for such a consideration was undertaken because of the fact that when obtaining electroadhesive joints, such small interelectrode gaps are used in them that it is no longer possible to consider the electric field to be uniform because of the discreteness of the electric charge. In addition, as shown by indirect experiments related to determining the strength (work of separation) of electroadhesive joints, depending on the various parameters of the technological process, the classical model based on the concept of “uniformly charged plane” gives underestimated values of electroadhesive forces. When developing a more accurate model, we will take into account that the length of the wave function of the electron itself in the material usually corresponds to not more than the units of nanometers, i.e., electrons can be roughly considered point charges.

The formula of a plane capacitor is usually used to calculate the interaction force caused by the attraction of opposite charges, distributed on the planes of the DEL:(1)$$p=\frac{\sigma^{2}}{2\varepsilon_{0}\varepsilon }=\frac{1}{2}\varepsilon_{0}\varepsilon E^{2}$$
where *p* is the ponderomotive pressure, $\varepsilon_{0}$—dielectric constant, ε—relative dielectric permeability of the material between interacting layers, σ—DEL surface charge density, E—the electric field strength. This implies that the charges on planes are distributed continuously and uniformly. In a stricter approach, it becomes obvious that such model does not correspond to the facts, because electric charge is discrete (the elementary charge is *e* = 1.6 × 10^−19^ C) and, therefore, the electric field of point charges system is not homogeneous. At the same time, the model for the calculation of the ponderomotive effect of the electric field on the basis of Equation (1) is acceptable in the case when the distance between the planes of the DEL is much greater than the distances between the discretely spaced elementary charges. In the case when the distance between planes of DEL is commensurate with the distance between the discretely spaced elementary charges, e.g., during the formation of electric adhesion compounds, the heterogeneity of the electric field manifests itself [4,6]. Thus, the results obtained by Equation (1) will be inaccurate.

Let us find out, at what distance between planes the Equation (1) will not adequately reflect the real physical situation. As an example let us consider a uniformly charged plane with surface charge density σ = 10^−4^ C/m^2^. As it is known that the electric field strength created in a vacuum (ε=1) will be equal to
(2)$$E=\frac{\sigma }{2\varepsilon_{0}\varepsilon }$$

Calculations according to Equation (2) give the value $E=\frac{\sigma }{2\varepsilon_{0}\varepsilon }$ = 5.65 × 10^6^ V/m.

It is precisely such and large in order of magnitude of the electric field strength that are used to obtain electroadhesive joints. In this way, the amount of elementary charges per 1 m^2^ area that obviously corresponds to the specified value of electric field strength is
(3)$$N=\frac{\sigma }{e}$$

Calculations according to Equation (3) give the value $N=\frac{\sigma }{e}$ = 6.25 × 10^14^ m^−2^. Consequently, the average distance between them is
(4)$$a=\frac{1}{\sqrt{N}}$$

Calculations according to Equation (4) give the value $a=\frac{1}{\sqrt{N}}$ = 4 × 10^−8^ m.

Thus, at the distance between the DEL planes corresponding to the order of tenths of a micron (i.e., micro- and nano-sizes), the Equation (1) can give untrue value. Note that it is the abovementioned distances between the charges that are used in the process of obtaining electroadhesive joints, since the gap between the parts to be joined is mainly due to the roughness of the surfaces to be joined, which are usually processed no worse than in the 11th class of cleanliness. Moreover, in the process of obtaining electric adhesion compounds surfaces are closed together and the distance between them decreases. For reference, we note that the value of the arithmetic deviation of the rough surface profile for items treated by the 11th class is R_a_ = 0.08 μm, and for treated by 14th class—R_a_ = 0.01 μm.

Thus, at small distances between the charged planes the discreteness of electric charge distribution should be taken into consideration. Indeed, in experiments [3,6,7] ponderomotive forces are often observed that have a much larger value than calculated according to Equation (1). One of the reasons for this is probably a discrete charge distribution.

### 2.2. Calculation of the Electric Field Strength Created by Charges Located on the Plane

It is known that with a continuous uniform distribution of electric charge over the plane, the strength of the electric field created is
(5)$$E_{c}=\frac{\sigma }{2\varepsilon_{0}\varepsilon }$$

For definiteness, let us add the index “*c*”: $E_{c}$—electric field strength for continuous charge distribution. Now let us calculate the electric field strength in the system, partially shown in Figure 1.

This is an infinite plane with discontinuous distribution of charges and with the same as in Equation (5) area-averaged surface charge density
(6)$$\sigma =\frac{q_{1}}{a^{2}}$$
where $q_{1}$ is the discrete charges shown by bold dots in Figure 1. In the case under consideration $q_{1}=e$ is the elementary charge. However, it should be noted that $q_{1}$ can also be understood as macroscopic charges unevenly distributed over the plane. Such charges, for example, are deliberately created in the so-called matrix electrets and, therefore, the approach considered below can be used, for example, to calculate the electric fields of matrix electrets and to study the effect of spatially separated charges in various capacitive systems, structures with a macroscopically distributed charge in the form of ionized impurities, quantum dots, etc., [7,8,9,10,11,12,13,14,15,16,17,18,19,20,21,22,23,24].

We define the field at a point M lying on the dotted line. Point M is chosen to consider the plane lying opposite one of the charges, because the charges of opposite signs of the DEL are located not chaotically, but correlated—against each other. To calculate the strength of the electric field $E_{d}$, taking into account the discreteness of the localization of the charge on the basis of the Coulomb law and the principle of superposition of fields, one can write
(7)Ed=∑n=0∞∑m=1∞q14πε0ε4rr2+a2m2+n232+q14πε0εr2=∑n=0∞∑m=1∞q1πε0εrr2+q1σm2+n232+q14πε0εr2.

The representation of Equation (7) in the form of a double series is due to the symmetry of the distribution of charges in each of the quarters of the plane relative to the charge onto which the point M is projected, the latter is represented in the formula by a separate term [6]. The summed elements are normal projections of the electric fields strengths created by each point electric charge, and are determined by the product of the modulus of this vector by the cosine of the angle between the vector and the normal to the plane. Tangential components are mutually compensated and are not considered in the calculations.

A comparison of the obtained values of the electric field strengths at different charge densities is graphically presented in Figure 2 and Figure 3 in order to find out the critical value of the distance to the plane *r_cr_*, at which the values of $E_{d}$ and $E_{c}$ differ insignificantly. Thus, it is possible to determine the limits of applicability of classical formulas for calculating the dependence of the strength of the electrostatic field on the distance to a charged object. As seen from Figure 2 and Figure 3 and the values indicated in the figure caption, as expected, the criterion that allows the classical approach to be applied in calculating electric fields is the relation *r_cr_* ≥ *a*, which with the actually used values of *σ* corresponds to submicron dimensions. At the same time, as *r* approaches zero, the value of $E_{d}$ increases indefinitely $E_{d}\sim \frac{1}{r^{2}}$. Here, one should take into account the termination of the implementation of the Coulomb’s law in the classical form at distances at which charges cannot be considered material points, therefore, the last expression can be used only for distances significantly exceeding the atomic dimensions and is limited, probably to *r >* 10^−9^ m. At distances much smaller, the complex nature of the force interatomic interaction begins to manifest itself—to the point that at very small distances the atoms will repel and our calculation becomes meaningless.

### 2.3. Calculation of the Electric Field Strength Created by Charges Located in a Layer of Finite Thickness

Consider the question of the location of elementary charges in a layer of finite thickness, consisting of several planes (Figure 4). To objectively assess the difference from the situation when the charge is on a plane, let us set the same total charge per unit volume as in the case considered above per unit area. The layer thickness is equal to *l⋅a*, where *l* is the number of planes, and *a* is the minimum distance between charges.

Let’s calculate the strength of the electrostatic field in the system shown in Figure 4. Similarly to the model with a charge distribution in one plane, we define the field at point M. The summed elements are normal projections of the intensity created by each point charge. Tangential components are mutually compensated and are not considered in the calculations. Summation over the variables *m* and *n* gives the total field from one plane, summation over *l*—from *L* planes.
(8)Ed=∑l=0L−1∑n=0∞∑m=1∞q14πε0ε4z+al(z+al)2+a2m2+n232+q14πε0ε(z+al)2

Numerical calculations show the insignificance of the influence that the layer thickness has on the resulting electric field. Therefore, it is enough to consider the case of only one layer (*L* = 1) and therefore for the electric field strength created by discrete charges the expression Equation (2) with a not very large number of layers and not too close a distance to the observation point can be used correctly.

## 3. Practical Findings

Direct use of Equations (7) or (8) leads to the computational problems, therefore, for the practical use of the obtained expression, it must be approximated by an analytical formula—which would deviate least from the value of the sum of the series under consideration. Let us start by considering the sum over the variable *n* (it is important to note that the order of the sums over *n* and over *m* is not important here because of the continuity of the integrand). Numerical calculations have shown that series Equations (7) and (8) converge slowly, i.e., the maximum values of *n* and *m* should be taken large enough. In this case, the contribution of each subsequent term with an increase in *n* and *m* is very small compared to the current value of the sum Equations (7) or (8). This gives reason to replace the summation by integration. The corresponding integral was analyzed when extending the domain of the function from the set of integers to a continuous positive semiaxis. The result of the corresponding calculations is a simple function that allows you to accurately calculate the fields of discretely distributed charges:(9)$$E_{d}\approx \frac{2}{\pi }\left[arctg\left(\frac{r\sqrt{\sigma }}{\sqrt{q_{1}}}\right)+\frac{\pi \sqrt{q_{1}}}{2r\sqrt{\sigma }}\right]E_{c}$$
where $E_{c}$ is calculated by the Equation (5), and *r* is the distance from the layer containing discretely distributed charges to the point at which the electric field strength is determined.

Let us analyze the resulting expression Equation (9). First consider the situation when the charges are very dense, i.e., σ→∞ and the plane can be considered uniformly charged. Then from Equation (9) it follows that $E_{d}\approx E_{c}$, which is quite expected. Now let us consider the case when there are very few charges on the plane; in fact, there is only one point charge. In this case σ→0 and from Equation (9), taking into account Equation (6), it follows that $E_{d}\approx \frac{a}{r}E_{c}$, where *a* is the distance between the charges. From what we have obtained, it can be seen that as r→0 we have $E_{\partial }\to \infty$, which is also expected, since when approaching a point charge, it is the electric field strength that increases indefinitely.

Thus, based on Equation (9), we conclude that to take into account the discreteness of the charge distribution, we can use the simple formula
(10)$$E_{d}\approx kE_{c}$$
where the coefficient *k* depends on the area-averaged surface charge density *σ* and the distance r from the charged plane to the observation point: (11)$$k=arctg\left(\frac{r\sqrt{\sigma }}{\sqrt{q_{1}}}\right)+\frac{\pi \sqrt{q_{1}}}{2r\sqrt{\sigma }}$$

Below is Table 1, in which the values of the coefficient k are calculated for various surface charge densities σ and distances r from the charged plane to the observation point.

The data in Table 1 need clarification: these data should not be taken literally as if, for example, the correct value of the electric field strength at σ = 10^−6^ C/m^2^ and r = 10^−9^ m differs from the traditionally calculated value by 400 times. In this regard, we recall that our calculation was carried out for the point M, which lies strictly opposite one of the charges that create the electric field. Moreover, for this situation, the calculation is fair. It is the presence of such large local electric fields and, accordingly, local forces that, in our opinion, explains, in particular, the abovementioned ability of the double electric layer to neutralize the embryonic cracks, formed in the plane of adhesive contact and subject to expansion under external influence. The calculated value of the electric field strength should not be confused with a certain average field of the considered plane with discrete charges. Indeed, based on Equation (6), it is easy to calculate that in this example, the distance between the charges will be *a* = 4 × 10^−7^ m. Thus, the height of the considered point M above the plane, Figure 1, is 100 times less than the distance between the charges. It is clear that in this case the field strength will strongly depend on the coordinates x and y, at which the vertical component of the electric field is determined. Therefore, the value given in the table essentially denotes the maximum value of the field strength that is locally observed above it. Over the greater part of the plane, the field strength in this example will be much less, and at the points between the charges, it may be negligible. Therefore, when calculating, for example, the integral force action of a field of such a plane, it is necessary to take into account the nature of the distribution of the electric field depending on the coordinates *x* and *y*. This is a rather difficult task and is not considered in this work. Therefore, it should be emphasized that if we understand by $E_{d}$ in Equation (9) some estimate of the average value of the electric field strength over a discretely charged plane, then the limits of applicability of this formula should be limited to values r≥a.

In this case, the position of the point M above the plane (in the sense of the values of its coordinates *x* and *y*) will not be decisive. In this regard, it has already been noted that series Equation (7) converges slowly, which physically means that in this case the field strength is determined mainly by the cumulative effect of the set of charges on the plane, and not by the specific nearest charge. Thus, at r≪a in Equation (7), the last term $\frac{q_{1}}{4\pi \varepsilon_{0}\varepsilon r^{2}}$—the field strength of the point charge, is of decisive importance, and at r≥a—the sum of the series is of decisive importance.

Calculations show that the consideration of charge discreteness leads to significant amendments in the direction of increased values of field strength. This also allows one to expect a more noticeable force action of the electric field in comparison with the classical calculation, at least in local areas. Additionally, it should be pointed out that with a discrete charge distribution, the electric field is highly inhomogeneous, and the inhomogeneous electric field has a pulling effect because of the gradient forces even on uncharged objects. All this can enhance the electroadhesive effects.

In Equation (9), it is assumed that the surface charge density σ can be of any nature, for example, chemical. When making electroadhesive connections, charges are created by using an external electrical voltage. In this case, charges of opposite signs, created by an external source, can be considered as charges of a flat capacitor and therefore can be written:(12)$$E_{c}=\frac{\sigma }{\varepsilon_{0}\varepsilon }$$

Expressing σ from Equation (12) and substituting into Equation (9), we obtain a formula connecting $E_{d}$ and $E_{c}$ for the case when an electric field is created by applying an electric voltage to the interelectrode gap:(13)$$E_{d}\approx \frac{2}{\pi }\left[arctg\left(\frac{r\sqrt{\varepsilon_{0}\varepsilon E_{c}}}{\sqrt{q_{1}}}\right)+\frac{\pi \sqrt{q_{1}}}{2r\sqrt{\varepsilon_{0}\varepsilon E_{c}}}\right]E_{c}$$

Table 2 for the case of nanometer (r = 10^−9^ m) and submicron (r = 10^−7^ m) dimensions of the gap shows the values of the electric field strengths $E_{c}$ and $E_{d}$. Recall that all calculations are performed under the assumption that the total charge of the plane is the same, regardless of the nature of the charge distribution over the plane. Additionally, the corresponding values of the surface charge densities σ calculated on the basis of Equation (12) and the distances between discrete charges *a* calculated on the basis of Equation (6) are given.

We emphasize once again that for r≥a, under the values $E_{d}$ presented in Table 2, one can understand some estimate of the average value of the electric field strength over a discretely charged plane. At r≪a, the calculated values $E_{d}$ should be considered as local values of the electric field strength in the vicinity of each discrete charge. It can be seen from the data obtained that for nano- and micro-sized interelectrode gaps, local electric fields can be orders of magnitude higher than the field strengths calculated from the average (measured) value of the surface density of the surface charge.

Considering that the force (ponderomotive) action of the electric field is proportional to the square of the electric field strength, the effect of the discreteness of the charge distribution can in this sense be very significant even with a relatively small value of the charge itself. The following simplified calculation illustrates this idea. Let us consider the situation when the charges on the electrodes are far enough from each other and therefore we can assume that the interaction occurs only in pairs between charges opposite to the electrodes, Figure 5.

Let us calculate for this case the ponderomotive pressure arising between the electrodes and causing their attraction. To do this, we take into account the Coulomb’s law, as well as the fact that for each pair of attracting charges there is an area *a*^2^. Then the corresponding pressure $p_{d}$ for a discrete charge distribution will be
(14)$$p_{d}=\frac{q_{1}^{2}}{4\pi \varepsilon_{0}\varepsilon r^{2}a^{2}}$$

The ponderomotive pressure $p_{c}$, which occurs when the same amount of charge is evenly distributed over the electrodes, can be determined based on Equations (1) and (6):(15)$$p_{c}=\frac{q_{1}^{2}}{2\varepsilon_{0}\varepsilon a^{4}}$$

Let us find out how significant an increase in pressure can be due to the discreteness of the charge distribution with a constant value of the total number of charges on the electrodes. For this, consider the ratio of $p_{d}$ to $p_{c}$, which we denote by *K*. From Equations (14) and (15) we obtain:(16)$$K=\frac{p_{d}}{p_{c}}=\frac{a^{2}}{2\pi r^{2}}$$

From Equation (16) it is seen that for sufficiently small r, i.e., when the electrodes are located close, we get K>1, therefore, with small interelectrode gaps, the effect of increasing the force action of the electric field will be observed in comparison with the classical model of a flat capacitor. The corresponding critical interelectrode distance $r_{c}$ at which the indicated effect begins to be observed can be calculated by equating Equations (14) and (15).
(17)$$r_{c}=\frac{a}{\sqrt{2\pi }}$$
or, taking into account (6)
(18)$$r_{c}=\sqrt{\frac{q_{1}}{2\pi \sigma }}$$

Table 3 presents the calculation data based on Equation (18).

The obtained values correspond to the nano- and micro-dimensions of the interelectrode gap.

Let us quantify the indicated effect of pressure increase. For this, we calculate the coefficient K based on Equations (6) and (16).
(19)$$K=\frac{p_{d}}{p_{c}}=\frac{q_{1}}{2\pi r^{2}\sigma }$$

From what has been considered, it is clear that the calculation must be performed at values $r\leq r_{c}$. As an example, we will choose the value σ = 10^−4^ C/m^2^ typical for the use of an electric field (this value, in accordance with Equation (12), corresponds to a field strength of ≈10^7^ V/m), and the thickness of the interelectrode gap will be r = 10^−8^ m. In this way, as can be seen from Table 3, $r\leq r_{c}$. Then, based on Equation (19), we get K = 2.55.

Calculations show that the effect under consideration for small sizes of interelectrode gaps (in comparison with the distance between charges on them) can be very significant and lead to an increase in pressure by orders of magnitude. It should be noted, however, that at nanometer sizes of interelectrode gaps, various quantum phenomena (primarily tunneling) can be observed, which can, to a certain extent, complicate the manifestation of the above effect. In particular, as the size of the interelectrode gap tends to be zero, the length of the electron wave function may be comparable with the size of the gap. Obviously, in this case, you cannot use the Coulomb law.

## 4. Materials and Experimental Methods

Direct experimental verification on the equipment available to us turned out to be difficult. Therefore, we used a large amount of our experimental results accumulated in the study of electroadhesive joints. From the most general concepts of adhesion, it follows that it is possible to regulate the adhesion by applying an electric voltage to the objects to be connected. Despite the fact that this possibility has been proven experimentally, it seems that currently the corresponding techniques are not widely used.

In the formation of electroadhesive joints, as already noted, the force action of an electric field in micro- and nanoscale interelectrode gaps is used. It is possible to achieve large values of electric field strengths, in particular, because of the migration polarization of the dielectric, “glued” to the metal or semiconductor. In this case, the charge is accumulated in a thin dielectric layer near the anode.

The scheme for manufacturing electroadhesive joints is shown in Figure 6.

In a number of our works [5,6] both the theoretical foundations of obtaining electroadhesive “glues” and some possibilities of their practical use were considered. In particular, it was found that the strength of the ionic dielectric-metal electroadhesive joints is at the level of the strength of similar joints obtained by diffusion welding. It was also shown that, based on the use of electroadhesive technology, it is possible to significantly increase the adhesion of films applied to substrates both after and during their deposition. Besides, considering that the field is heterogeneous, due to the gradient forces, it may cause attraction to even electrically neutral objects, e.g., atoms sprayed on the substrate.

To quantify adhesion, the scratch method is often used [23,24]. However, in our experiments, it turned out to be unacceptable because of the relatively large thickness of the aluminum film connected by the electroadhesive method to the dielectric substrate. Therefore, we used the Krotova and Deryagin adhesion tester (Figure 7) [4], which we adapted to measure the adhesion of various coatings: if the film was a metal foil, then the load was attached directly to the foil; if the film was deposited on the surface by, for example, thermal evaporation in a vacuum, then a thin adhesive tape (scotch tape) was glued to the deposited film coating. Because of the strong adhesion of the adhesive layer of the tape to the investigated electrically conductive film when it was torn off by a load (by changing the angle, Figure 7), with not very high adhesion, peeling occurred along the film–substrate interface, which made it possible to quantitatively judge its adhesion to the substrate by the size of the separation angle.

In this case, the work of tearing off the film (work of separation) is found by the expression [4]
(20)$$A=\frac{mg}{b}\left(1-cos\alpha \right)$$
where *m* is the mass of the load, *g* is the acceleration of gravity, *b* is the width of the film being torn off, α is the measured angle at which the torn off begins.

With high adhesion of the film to the substrate, only partial peeling of the film from the substrate occurred during measurements. In this case, according to the results of visual observations, the relative fraction of the exfoliated area of the film $\frac{S_{s}}{S_{0}}$ was calculated and the work of separation calculated according to Equation (20) was divided by this coefficient. Because of the fact that the value $\frac{S_{s}}{S_{0}}$ < 1, the value of the work of separation was greater than according to Equation (20). Thus, the work of separation was determined by the formula
(21)$$A=\frac{mg}{b}\left(1-cos\alpha \right)\frac{S_{0}}{S_{s}}$$

Of course, the proposed approach is not rigorous and was used only because it makes it relatively easy to get a quantitative idea of the adhesion of the film to the substrate, when it exceeds the adhesion of the scotch tape to the film in a part of the area or the strength of the thin foil used as a coating. There are other methods for determining adhesion, in particular, various modifications of the scratching method are widespread [23,24]. These methods can be used in the case of thin coatings that make it possible to observe their peeling under the action of a mechanical force applied to the tip. As a rule, such coatings are obtained using technologies that do not involve a gap between the substrate and the coating, for example, vacuum spraying. In our case, it is of interest to trace the influence of electric fields precisely in the presence of this gap. For this reason, we are considering here the application of relatively thick film materials (aluminum foil) onto the substrates (glass) by the electroadhesive method. In this case, at the beginning of the joining process, there will be a thin gap between the parts to be joined due to the roughness and waviness of the surfaces to be joined. To assess the adhesion achieved in this case, the above method of Deryagin and Krotova is more appropriate. In addition, this method, in contrast to other methods of quantitative determination of adhesion, does not require special expensive equipment [9,21]. Also, earlier we developed a model for calculating the ponderomotive pressure and the area of actual contact, taking into account the roughness of one of the contact surfaces [25], Figure 8.

The scope of this work does not allow us to consider this model in detail. The presentation of this technique is the subject of a separate article. Therefore, we will restrict ourselves here to the most simplified, but containing the basic ideas about the issue under consideration. We point out that the relative fraction of the actual contact area, obviously proportional to the strength of the resulting joints (section 3 in Figure 8), was determined on the basis of the expression [25]:(22)$$\frac{S_{a}}{S_{0}}=1-exp[-\frac{\bar{p}}{p_{0}}]$$

Here $S_{0}$ is the total (geometric) contact area, $S_{a}$ is the actual contact area, $p_{0}$ is a coefficient proportional to the Young’s modulus of the material being deformed and also depending on the surface roughness, $\bar{p}$ is the area-average pressure caused in our case by the action of electric fields in the air gap and in the areas of actual contact, Figure 9. The physical meaning of $p_{0}$ corresponds to the value of the pressure $\bar{p}$ averaged over the area, at which 63% of the nominal area are in actual contact. The $\bar{p}$ value was calculated on the basis of the formulas for the ponderomotive action of an electric field without taking into account the discreteness of the charge in this system [25].

In the process of electroadhesive bonding (in the considered experiments, aluminum foil with window glass was connected) for some time (usually tens of minutes at an elevated temperature), a charge accumulates in the region $x_{m}$, Figure 8. As shown by our earlier calculations confirmed by experiment, a typical value of the thickness of this region is a value of several microns [6,25]. Therefore, for further estimates, we take $x_{m}$ = 5 μm. In fact, the thickness of the layer near the anode charge localization in the dielectric increases with increasing voltage. However, this increase is not sharp (as calculations show, it is proportional to the square root of the voltage). In addition, the thickness of this layer, at almost any reasonable electrical voltage, is noticeably greater than the thickness of the air gap. Let us take the initial thickness of the air gap at the level of roughness, which was *R_z_* = 0.04 μm. Because of the fact that the thickness of the region $x_{m}$ is much greater than the thickness of the air gap d, the pulling field strength $E_{1}$ in the classical version weakly depends on the thickness of this gap:(23)$$E_{1}=\frac{\varepsilon_{2}U_{0}}{\varepsilon_{1}x_{m}+\varepsilon_{2}\bar{d}}\approx \frac{\varepsilon_{2}U_{0}}{\varepsilon_{1}x_{m}}$$
where $\varepsilon_{1}$ = 1 is the dielectric constant of air, $\varepsilon_{2}$ = 4.5 is the dielectric constant of glass. $U_{0}$ is the electrical voltage applied to the materials to be joined. It is due to the presence of an electric field $E_{1}$ in the air gap that the deformation of microprotrusions and an increase in the area of actual contact occurs, which leads to the “gluing” of the materials to be joined. Therefore, as an estimate of the pressure $\bar{p}$, we take the pressure $p_{1}$ created by the field $E_{1}$.

We have carried out numerous experiments to measure the strength (work of separation) of various electroadhesive “glues” depending on the various technological factors—the temperature of the joint, the roughness of the surfaces to be joined, the holding time, etc. All of them confirm the decisive influence of the electric field on the strength of the joints.

Obviously, the most direct factor affecting the magnitude of the electric field is the electric voltage, Figure 9.

It was noticed that at low voltage values and, consequently, low surface charge densities created by it, the calculated dependence (curve 1 in Figure 9) of the relative fraction of the actual contact correlates worse with the measured bond strength than in the case of high electric voltages. It seems that this result indirectly confirms the ideas developed in this work. Indeed, from the above it follows that at high densities of surface charges and interelectrode gaps of submicron size, there is practically no effect of increasing the force action of the electric field due to the discreteness of the charge distribution (see Table 2 and Table 3). At relatively low electrical voltages, this effect, taking into account the small values of the size of the localization of the field (in the air gap—on the order of and less than *R_z_* = 0.04 μm—the roughness of the surfaces to be joined), as shown above, can turn out to be significant. When plotting curve 2, Figure 9, this effect was taken into account in accordance with the methods discussed above, and, as can be seen from Figure 9, the correlation between the calculated value of the relative area of actual contact and the strength of the connection is much better in a wide range of used electric voltages. In our opinion, this indirectly confirms the concepts considered in the work.

Let us explain in more detail the methods mentioned, which were used to construct the curve (2), Figure 9. Based on Equations (12), (19), and (23), it can be obtained that, due to the discreteness of the charge distribution in the air gap, an increase in the ponderomotive pressure in comparison with the classical calculation occurs by a factor of *K*:(24)$$K=\frac{p_{d}}{p_{c}}=\frac{q_{1}x_{m}}{2\pi r^{2}\varepsilon_{0}\varepsilon_{2}U_{0}}$$

Under *r*, as above, we mean the distance between the interacting charges. In our case, these are charges on the aluminum foil and in the region $x_{m}$ in the dielectric. The distance between these charges, because of the complex roughness profile, is different and varies from the maximum, as an estimate of which one can take the roughness *R_z_* = 0.04 μm, to the minimum, corresponding to interatomic distances. Thus, an accurate calculation is rather difficult, so we will restrict ourselves to only rough estimates. In this regard, Table 4 presents the values of the coefficient $K=\frac{p_{d}}{p_{c}}$ calculated on the basis of Equation (24) for various values of *r* from the specified range.

Some cells of Table 4 have dashes because the values calculated by Equation (24) are less than 1. Note that these cells of the table correspond to high voltages and, accordingly, high surface charge densities. In this case, it is no longer possible to use the simplified model shown in Figure 6, in which only the pairwise interaction of charges was taken into account. It was already noted above that Equation (14) and subsequent formulas, logically connected with it, correspond to a situation when the charges on the electrodes are far enough from each other and therefore it can be assumed that the interaction occurs only in pairs between charges opposite to the electrodes. At high surface charge densities, it is already necessary to use more exact Equations (7) or (9). At high surface charge densities, in this case, as can be seen from Table 1 and Table 2, the difference between the classical calculation using the concept of a uniformly distributed charge and the calculation taking into account the discreteness of the charge disappears. Therefore, in reality, the coefficient *K* cannot be less than one.

Curve (2) in Figure 9 was constructed, taking into account what was considered, as follows. Instead of the $\bar{p}$ value, which we previously calculated classically, taking into account the accumulation of space charge in the dielectric and the presence of a certain pulling charge due to this, not only in the air gap, but also at the points of actual contact [6], in Equation (22) the value was substituted
(25)$$p_{d}=K\bar{p}$$

The value of the coefficient *K* was chosen for the case *r =* 4 × 10^−9^ m (Table 4). This choice is explained by the fact that when a strong connection is obtained, a significant increase in the area of actual contact should occur, and the thickness of the air gap should significantly decrease, for example, by an order of magnitude. In reality, this is exactly what happened—the connections were vacuum-tight, which was checked with a helium leak detector. No Newtonian rings were observed in the reflected light because of the possibility of a residual air gap after joining.

It should be noted that when manufacturing an electroadhesive joint, much more complex processes actually occur than were considered. Moreover, it is not obvious that the actual contact area and the measured bond strength are strictly proportional to each other. It was for this reason that there was no point in making an accurate calculation. Our goal was only to demonstrate experimentally the possible influence of discretely distributed charges.

## 5. Results and Discussion

Thus, at submicron distances to the charged plane, the magnitude of the electric field strength and its force effect can be significantly higher than that calculated by the classical formulas. The expressions obtained in the work make it possible to quantitatively evaluate the criteria for the need to switch to taking into account the discreteness of the charge.

With the help of the approach developed in the work, an explanation can be given for some experimental data, which are difficult to explain with the classical approach. For example, as a result of classical calculations when analyzing electric fields in most real situations, the values of electric field strengths are obtained, as a rule, no more than 10^7^ V/m. When using the classical approach, the corresponding pressures, according to Equation (1), will be of the order of 1 kPa. The adhesion strength of various joints can be at the level of the cohesive strength of the solids being joined, and the latter is of the order of 100 MPa. Thus, this approach does not show a significant contribution of the electrostatic component to the adhesive strength of the joint. At the same time, in the above literature data, there are numerous evidences of a significant influence and various noticeable manifestations of the electrostatic component of adhesion.

In our opinion, this situation can be easily explained using the approach developed in this work: the local values of the field strengths can be significantly higher than those indicated above, and for small interelectrode gaps, the force action of the electric field can significantly increase in comparison with the case when the discreteness of the charge is not taken into account. The considered features of the electric fields can, in particular, explain the strong influence of the electric field on the adhesion of conducting films deposited on dielectric substrates.

Mathematical equations provided in this work are believed to be essential for calculating the various effects in MEMS—technologies where small distances and the electrostatic principle of controlling are often used. Besides, the developed method can be used at calculation of the electric effects resulting from not uniform distribution of a charge caused, for example, by structure defects.

It should also be noted that the approach developed in this work for calculating electric fields can also be used for the case of relatively large interelectrode gaps, when the source of the field is macroscopic regions unevenly charged over the surface. An example of such objects are the so-called matrix electrets [1].

## 6. Conclusions

The performed calculations have shown that at small distances to the charged plane, electric fields may appear that significantly exceed the values calculated by the classical formulas. The considered field with a discrete charge distribution is inhomogeneous; therefore, due to gradient forces, it can have a pulling force effect even on electrically neutral objects, for example, atoms deposited onto a substrate, and have other manifestations that can find practical applications. The performed calculations show that the considered effects become significant, as a rule, at submicron sizes of the interelectrode gaps. The expressions obtained in this work make it possible to quantitatively evaluate the influence of the discreteness of the charge distribution on the features of electric fields in small interelectrode gaps. In this case, the term “charge” does not necessarily mean the charge of an electron. These can be regions of relatively large dimensions on the surface of the material, in which, for one reason or another, there is an increased density of electric charge.

## Figures and Tables

**Figure 1 materials-13-05669-f001:**
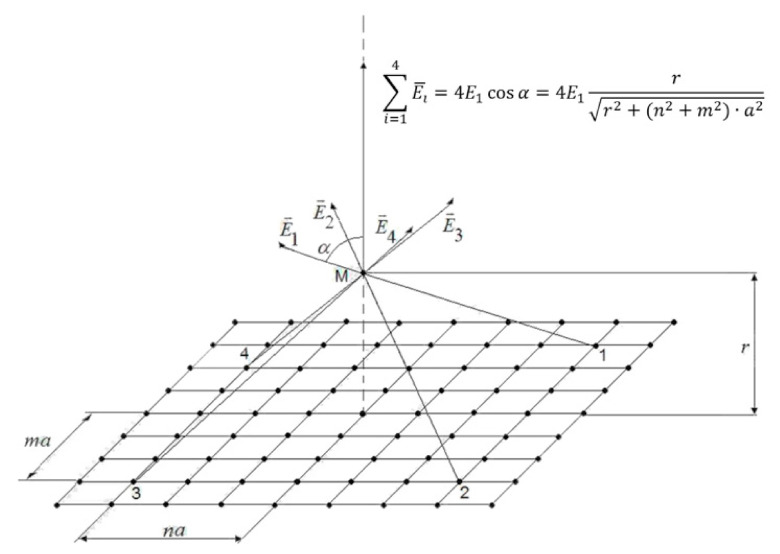
Model of a plane with discontinuous distribution of charges (*a*—is the distance between charges). As an explanation, several vectors of electric field strength and their resulting are shown.

**Figure 2 materials-13-05669-f002:**
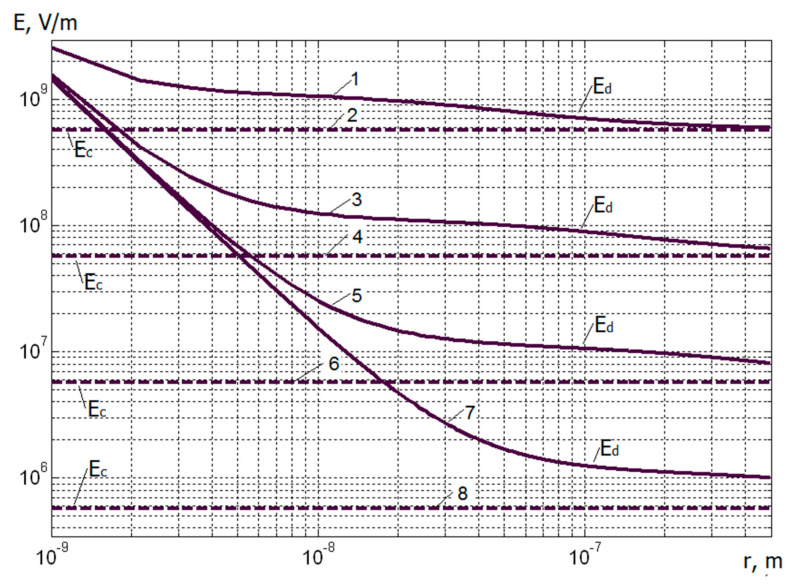
Dependences of the strengths of the electrostatic fields on the distance to the plane for various parameters. 1—$E_{d}$, *σ* = 10^−2^ C/m^2^; *a* = 4 nm; 2—$E_{c}$, *σ* = 10^−2^ C/m^2^; *a* = 4 nm; 3—$E_{d}$, *σ* = 10^−3^ C/m^2^; *a* = 0.04 μm; 4—$E_{c}$, *σ* = 10^−3^ C/m^2^; *a* = 0.04 μm; 5—$E_{d}$, *σ* = 10^−4^ C/m^2^; *a* = 0.13 μm; 6—$E_{c}$, *σ* = 10^−4^ C/m^2^; *a* = 0.13 μm; 7—$E_{d}$, *σ* = 10^−5^ C/m^2^; *a* = 0.4 μm; 8—$E_{c}$, *σ* = 10^−5^ C/m^2^; *a* = 0.4 μm.

**Figure 3 materials-13-05669-f003:**
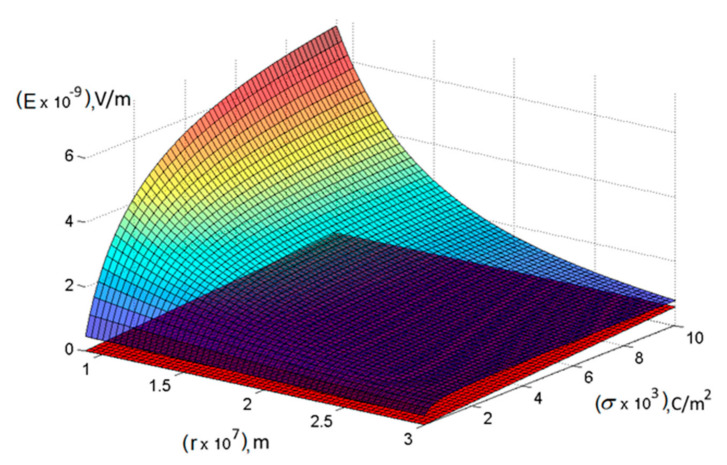
Dependences of the electrostatic fields strengths on the distance to the plane and the surface charge density (the red plane corresponds to a continuous charge distribution, a curved surface—to a discrete one).

**Figure 4 materials-13-05669-f004:**
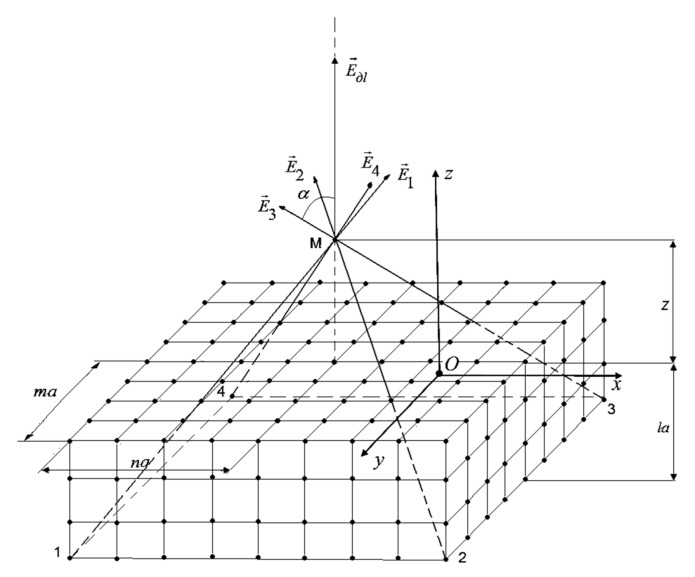
Layer model with intermittent charge distribution. Explanatorily depicts several vectors and their resultant electric field strength at a point located on one normal with the charge. *n*, *l*, and *m* are integers.

**Figure 5 materials-13-05669-f005:**
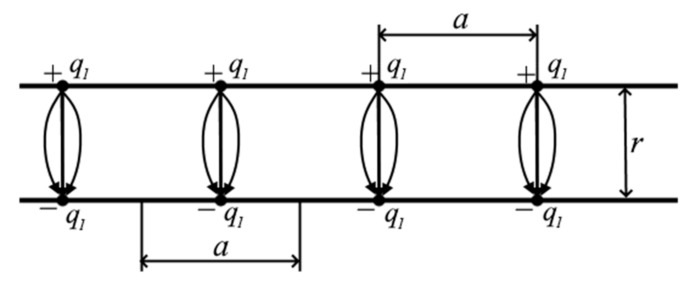
Pairwise interacting charges discretely distributed on the electrodes. $q_{1}$—discrete point charges charges on the electrodes, r—distance between electrodes, a—distance between charges on a plane.

**Figure 6 materials-13-05669-f006:**
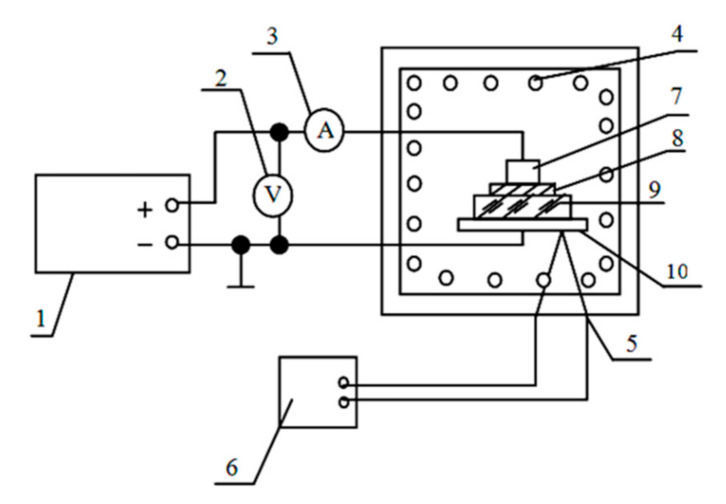
Installation diagram for manufacturing electroadhesive joints: 1—power supply, 2—voltmeter, 3—self-recording milliammeter, 4—muffle furnace, 5—thermocouple, 6—potentiometer, 7, 10—electrodes, 8—metal (semiconductor), 9—dielectric.

**Figure 7 materials-13-05669-f007:**
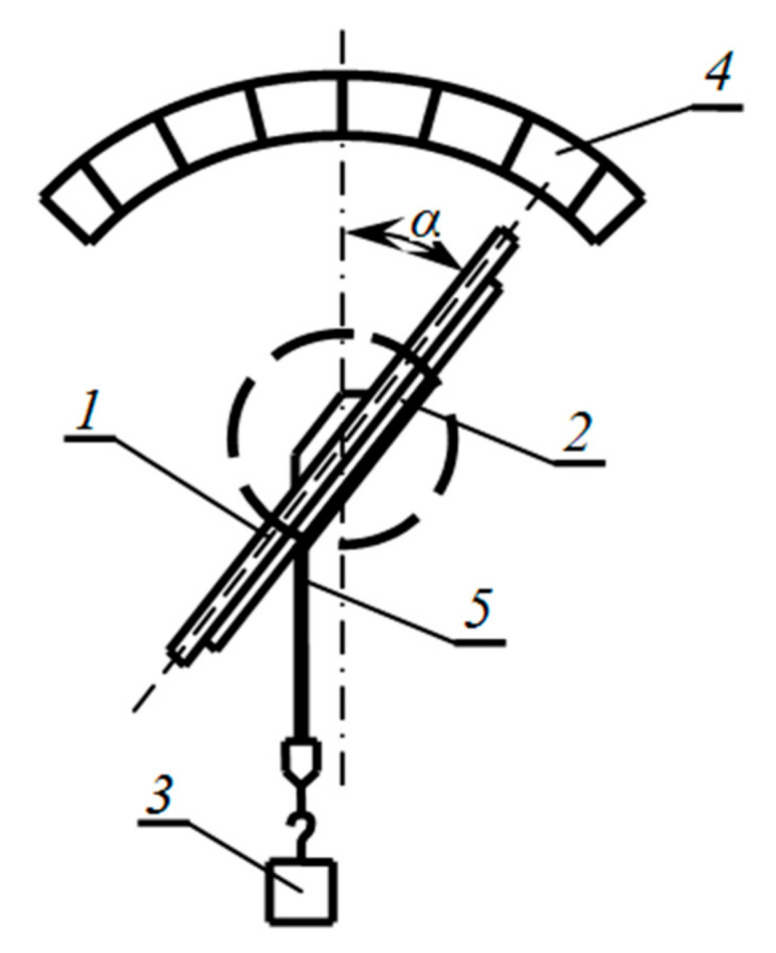
Angular adhesion meter by Krotova and Deryagin [4]. 1—Plate, 2—prototype (substrate), 3—weight, 4—goniometer scale, 5—film peeled off from the substrate.

**Figure 8 materials-13-05669-f008:**
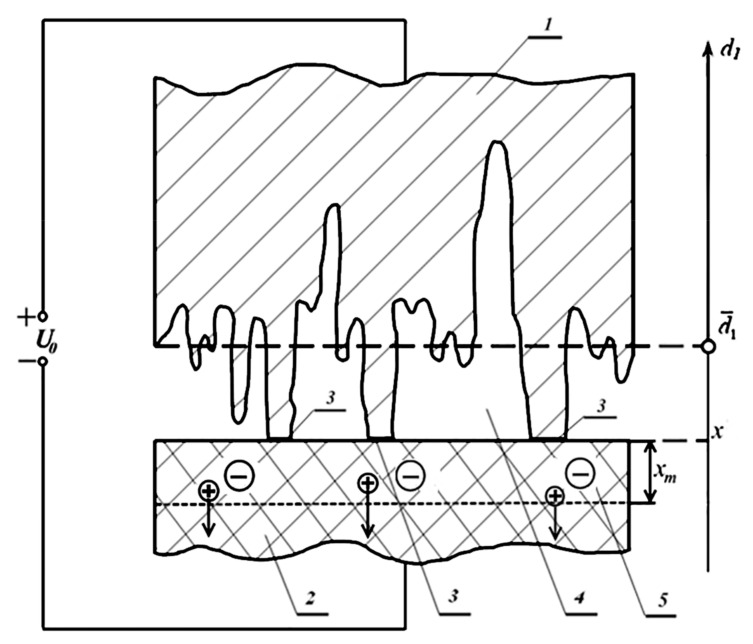
A model for calculating the ponderomotive pressure when manufacturing an electroadhesive joint ionic dielectric-to-conductor [25]. 1—Metal (semiconductor); 2—dielectric; 3—area of actual contact; 4—air gap due to the roughness of the contacting surfaces; 5—layer of localization of space charge in the dielectric.

**Figure 9 materials-13-05669-f009:**
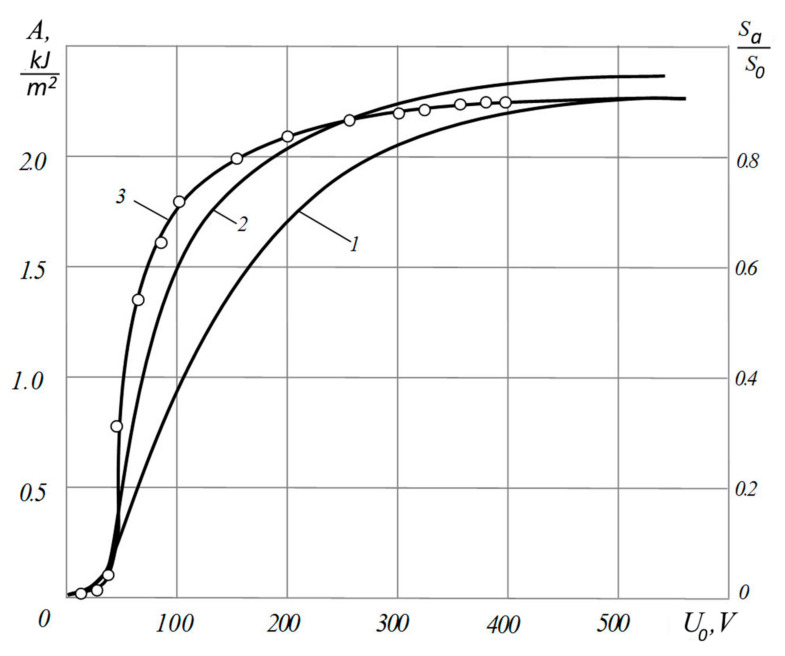
Experimental dependence of the strength of the electroadhesive joint aluminum foil-polished window glass (curve 3) and the calculated dependences of the actual contact relative area *S_a_*/*S_0_* on the electric voltage for the same pair of materials (curves 1, 2). 1—Calculation taking into account the accumulation of space charge, in the anode region of the dielectric; 2—calculation taking into account the accumulation of space charge in the anode region of the dielectric and taking into account the discreteness of the charge distribution. $\varepsilon_{2}$= 4.5; $p_{0}$ = 3.59 × 10^6^ Pa; $\bar{d_{1}}$ = *R_z_* = 0.04 μm.

**Table 1 materials-13-05669-t001:** Values of the coefficient $k=\frac{E_{d}}{E_{c},}$ for various surface charge densities σ and distances r from the charged plane to the observation point.

	*r*, m	10^−10^	10^−9^	10^−8^	10^−7^	10^−6^	10^−5^
*σ* C/m^2^							
10^−6^		4000	400	40.0	4.16	1.16	1.0145
10^−4^		400	40.0	4.16	1.16	1.0145	1.0015
10^−2^		40.0	4.16	1.16	1.0145	1.0015	1.0001

**Table 2 materials-13-05669-t002:** Strengths of electric fields $E_{c}$ and $E_{d}$ for the case of a nanometer-sized gap (r = 10^−9^ m) in vacuum (*ε* = 1). An electric field is created by applying an electric voltage to the interelectrode gap. Additionally, the corresponding values of the density of surface charges σ, and the distances a between discrete charges are given.

$E_{c},\;V/m$	10^2^	10^4^	10^6^	10^8^	10^10^
$E_{d}$, V/m (r = 10^−9^ m)	1.34 × 10^6^	1.34 × 10^7^	1.34 × 10^8^	1.35 × 10^9^	1.75 × 10^10^
$E_{d}$, V/m (r = 10^−7^ m)	1.34 × 10^4^	1.35 × 10^5^	1.75 × 10^6^	1.05 × 10^8^	1.0005 × 10^10^
σ, C/m^2^	8.85 × 10^−10^	8.85 × 10^−8^	8.85 × 10^−6^	8.85 × 10^−4^	8.85 × 10^−2^
a, m	1.34 × 10^−5^	1.34 × 10^−6^	1.34 × 10^−7^	1.34 × 10^−8^	1.34 × 10^−9^

**Table 3 materials-13-05669-t003:** The values of the critical distances between the electrodes calculated at various surface charge densities, at which the effect of an increase in the force action of the electric field begins to be observed in comparison with the classical model of a flat capacitor.

σ, C/m^2^	10^−8^	10^−6^	10^−4^	10^−2^
$r_{c}$ **, m**	1.596 × 10^−6^	1.596 × 10^−7^	1.596 × 10^−8^	1.596 × 10^−9^

**Table 4 materials-13-05669-t004:** Values of the coefficient $K=\frac{p_{d}}{p_{c}}$ at different values of the distances r between the interacting charges, corresponding to the roughness of the surfaces to be joined, at different applied electric voltages.

*U*_0_, V	1	10	100	1000
*K* (*r* = 4 × 10^−8^ m)	2.00	-	-	-
*K* (*r* = 10^−8^ m)	32.0	3.20	-	-
*K* (*r* = 4 × 10^−9^ m)	200	20	2	-
*K* (*r* = 10^−9^ m)	3200	320	32.0	3.20

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
