# Peer review of "Features of Electrostatic Fields and Their Force Action When Using Micro- and Nanosized Inter-Electrode Gaps"

_materials, 2020, doi:10.3390/ma13245669_

Round 1

Reviewer 2 Report

The paper is focused on the study of a mathematical model to consider the electric field in micro and nanosized gaps. In general, the scope of the work is good but there are some other issues that should be considered before publication. Please find below my comments and suggestions:

a) Some general comments regarding formatting. There are some aspects regarding on format that should be modified. For example, in the title there is a ‘typo’ in the word interelectrode ‘gapes’. The format of the article can be improved. For instance, the paragraph in line 59 begins with a reference in brackets [1] points out when usually, we cite the authors Derjagin and cols [1] or else Smilga and cols [1]. This applies to all document.

In the introduction the points 1 and 2 (in lines 60 and 65, respectively), probably it would be better to use a) and b) to avoid confusion with sections of the document. In other sections of the document, there are single lines or short paragraphs that maybe can be presented differently.

In some parts of the document, there are some too colloquial expressions. 

b) Abstract. It can be shortened (about 200 words, 220 max) and highlight the most important aspects such as the development of a mathematical and physical model for the study of an non-uniform electric field.

c) Please make a critical revision of the introduction. The presentation of the ideas is not clear. Also, the way references are presented is uneven. For example, in line 49 associated to that paragraph are cited references 4 – 24, which are almost all references of the introduction. Actually, no more references are added in the rest of the manuscript.

The last paragraph usually emphasizes the aim of this work.

d) In section 2. Materials and Methods. Probably, you should re-considered changing the title of this section. When reading the section, nothing is said about the ‘materials used’, perhaps just ‘Methodology’. Also, consider the possibility of adding sub-sections to favor the understanding of the methodology used. For example, section 2.1. section 2.2. etc…

In line 124, referring to roughness, which values of roughness are used (Ra, Rz, Sa,…). For example, in line 446, you refer to Rz = 0.04 micrometers whereas in line 124 it is not mentioned the kind of parameter used to refer to roughness.

e) Section 3. Results. In this section, we expect to find the results of the work, and there are two sub-sections 3.1. Practical findings, which are results of the work and section 3.2. Experimental Part. This section is usually within the “Materials and Methods” or if you wish at the end of the document, but is in the section of ‘Results’. You may separate the experimental ‘details’ and then include the results.

In Figure 8, you use a model proposed in a former work, please include your reference in Figure caption too, otherwise it seems it is something new from this research work.

There are a few self-citations, which if it is essential to cite them is correct, but please consider citing other works. “In a number of our works [2,3]”.

f) Discussion section can be improved. Also, I suggest adding a section of “conclusions” to highlight the findings of this work.

Finally, I would like to comment that the scope of the paper seems interesting to me and it can be for the readership, but the overall structure should be improved. Perhaps you may consider the possibility of using a section of ‘supplementary material’ so that you put the essential in the paper and the rest is provided separately.

Round 2

Reviewer 2 Report

Dear authors, 

Thank you for addressing most of my comments and suggestions in the revised version of the manuscript provided. I consider that the manuscript can be published. 

Kind regards.